# Predicting Remaining Survival of Glioblastoma Patients with Radiomics Analysis Based on ^18^F-DOPA PET Images

**DOI:** 10.3390/cancers17213560

**Published:** 2025-11-03

**Authors:** Jing Qian, Deanna Hasenauer, William G. Breen, Paul D. Brown, Christopher H. Hunt, Mark S. Jacobson, Derek R. Johnson, Timothy J. Kaufmann, Bradley J. Kemp, Sani H. Kizilbash, Val J. Lowe, Michael W. Ruff, Jann N. Sarkaria, Joon H. Uhm, Mark J. Zakhary, Maasa H. Seaberg, Hok Seum Wan Chan Tseung, Elizabeth S. Yan, Yan Zhang, Nadia N. Laack, Debra H. Brinkmann

**Affiliations:** 1Department of Radiation Oncology, Mayo Clinic, Rochester, MN 55905, USA; breen.william@mayo.edu (W.G.B.); brown.paul@mayo.edu (P.D.B.); sarkaria.jann@mayo.edu (J.N.S.); tseung.hok@mayo.edu (H.S.W.C.T.); yan.elizabeth@mayo.edu (E.S.Y.); zhang.yan@mayo.edu (Y.Z.); laack.nadia@mayo.edu (N.N.L.); 2Department of Radiation Oncology, Mayo Clinic, Jacksonville, FL 32224, USA; hasenauer.deanna@mayo.edu; 3Department of Radiology, Mayo Clinic, Rochester, MN 55905, USA; hunt.christopher@mayo.edu (C.H.H.); jacobson.mark17@mayo.edu (M.S.J.); johnson.derek1@mayo.edu (D.R.J.); kaufmann.timothy@mayo.edu (T.J.K.); kemp.brad@mayo.edu (B.J.K.); vlowe@mayo.edu (V.J.L.); 4Department of Medical Oncology, Mayo Clinic, Rochester, MN 55905, USA; kizilbash.sani@mayo.edu; 5Department of Neurology, Mayo Clinic, Rochester, MN 55905, USA; ruff.michael@mayo.edu; 6Department of Neurosurgery, Mayo Clinic, Rochester, MN 55905, USA; uhm.joon@mayo.edu; 7University of Florida Health Proton Therapy Institute, Jacksonville, FL 32206, USA; markzakhary@ufl.edu; 8Department of Radiation Oncology, University of California San Francisco Medical Center, San Francisco, CA 94143, USA; maasa.seaberg@ucsf.edu

**Keywords:** ^18^F-DOPA, amino-acid PET tracer, glioblastoma, post-treatment follow-up, radiomics, manifold learning, machine learning, remaining survival

## Abstract

**Simple Summary:**

Post-treatment surveillance of glioblastoma is challenging due to the complications of discriminating against treatment effects using conventional imaging. This study proposes to address the challenge by linking radiomics features from ^18^F-DOPA PET images with remaining survival in glioblastoma patients. Employing customized feature selection algorithms and manifold learning, quantitative features were extracted for machine learning, resulting in heightened sensitivity for detecting changes in survival risk. The proposed map for remaining survival has the potential to enhance personalized medicine. Overall, this research underscores the potential effectiveness of ^18^F-DOPA PET and machine learning models in forecasting survival outcomes and monitoring tumor progression among glioblastoma patients undergoing dose-escalated radiation therapy.

**Abstract:**

Background: Post-treatment prognosis and monitoring are critical for determining the timing of salvage treatment in glioblastoma patients but has been challenging due to difficulties differentiating progression from treatment effects in conventional images. This exploratory study aimed to establish the correlation of radiomics image features from time series of amino acid tracer ^18^F-DOPA PET images, with outcomes, using machine learning and dimension reduction analysis. Methods: ^18^F-DOPA PET images were collected for a patient cohort with wild-type IDH and unmethylated MGMT who underwent dose-escalated radiation therapy. Quantitative features were derived from the high uptake region (T/N > 2.0) in pre- and post-radiation therapy follow-up ^18^F-DOPA PET images. A customized workflow was utilized for pre-selecting predictive features, followed by manifold learning. Machine learning algorithms were employed to establish associations between imaging features and remaining survival (RS), defined as the time between a follow-up scan and date of death. Results: The ML models exhibited 81–83% ROC_AUC in predicting RS evaluated on an independent test dataset. A RS map is proposed for monitoring tumor alterations through serial ^18^F-DOPA PET scans, demonstrating superior sensitivity and better correlation with survival compared to the RANO criteria. Conclusions: Our study demonstrates that ML models utilizing FU ^18^F-DOPA PET images have the potential to effectively predict future survival outcomes in patients with glioblastoma treated with dose-escalated radiation therapy. The capability to assess changes in tumor over time through imaging can potentially assist in patient stratification and the selection of salvage treatments, while also aiding in distinguishing treatment effects from genuine tumor progression.

## 1. Introduction

Glioblastoma [1] is the most common primary malignant brain tumor with poor prognosis. It is typically treated with surgical resection followed by concurrent radiotherapy and chemotherapy [2], sometimes supplemented with tumor treating fields [3], and then adjuvant temozolomide (TMZ). Due to the aggressive nature of glioblastoma, despite these multimodality treatments, the median overall survival is less than 2 years after diagnosis [4]. Salvage treatment is commonly administered upon disease recurrence post-treatment to extend patient survival [5,6]. However, accurately pinpointing recurrence, whether through radiographic or clinical means, poses a considerable challenge due to the complexities arising from the effects of multimodal therapy.

Following treatment for glioblastoma, treatment response surveillance imaging is performed using standard T1-weighted (with and without contrast) and T2-weighted magnetic resonance imaging (MRI) sequences. Unfortunately, despite modern imaging assessment tools, it remains difficult to differentiate tumor progression from treatment effect [7,8], even with established response assessment criteria such as the response assessment in neuro-oncology (RANO) criteria [9,10]. The interaction between radiotherapy and chemotherapy can enhance contrast uptake through various mechanisms, for example, through a disruption of the blood–brain barrier and/or through an alteration in cell metabolism [11], potentially leading to contrast enhancement on imaging with image features similar to those related to true progressive disease (PD), falsely suggesting tumor progression. This phenomenon, known as pseudoprogression (PsP), occurs with reported incidence rates ranging from 9% to 32% [12,13,14]. An incorrect interpretation of progression may either delay time-sensitive salvage treatment or interrupt the effective primary treatment, both resulting in suboptimal tumor control. This difficulty of differentiating treatment effect from tumor progression may become more prevalent with advancements in treatment options that may increase radiographic treatment effect [15,16], such as radiation dose escalation, radiosensitizers, immunotherapy, and other systemic therapies. Consequently, there is a critical need for improved methods to differentiate treatment effect from tumor progression.

In recent years, advanced imaging techniques have emerged as valuable tools for post-treatment surveillance. Examples include advanced MRI modalities such as perfusion and diffusion MRI [17,18,19,20], as well as positron emission tomography (PET) utilizing amino acid tracers [21,22,23,24]. These methods offer functional and metabolic data that can enhance traditional anatomical imaging, aiding in the differentiation of treatment effects from tumor cell proliferation. Studies [25,26,27,28] have suggested PET with amino-acid tracers can provide more specific uptake in tumor tissue than in areas of radiation-induced normal tissue response. Moreover, there have been significant advancements in image analysis techniques. Complex modeling approaches like radiomics [29,30,31,32] and deep learning (DL) [33,34] are now commonly employed for extracting image features and predicting prognosis.

In this exploratory study, we utilize radiomics and machine learning, including nonlinear dimension reduction techniques, to study a time series of PET images with the amino acid tracer 3,4-dihydroxy-6-^18^F-fluoro-L-phenylalanine (^18^F-DOPA), on a cohort of newly glioblastoma patients treated with escalated radiation dose [16]. Although potential survival benefits have been demonstrated in the trial with ^18^F-DOPA PET-guided escalated dose, the disconnect between progression-free survival (PFS) per conventional MR and overall survival (OS) of the patients suggests a high incidence rate of PsP, making the standard of care post-treatment surveillance difficult. This study focuses on the cohort of patients with poorer prognosis (wild isocitrate dehydrogenase (IDH) and unmethylated O^6^-methylguanine–DNA methyltransferase (MGMT) status). As an exploratory and hypothesis-generating study limited by the data size, our aim is to predict remaining survival after each follow-up (FU) image, thereby demonstrating the potential value of ^18^F-DOPA as a prognostic tool and a potential aid in differentiating treatment effect from tumor progression.

## 2. Materials and Methods

### 2.1. Patients and ^18^F-DOPA PET Surveillance Images

This study analyzed the ^18^F-DOPA PET surveillance images from patients enrolled in an Institutional Review Board (IRB)-approved prospective trial (NCT01991977). All participants received ^18^F-DOPA-guided dose-escalated radiation therapy (DERT). Inclusion criteria in this secondary analysis required patients to have newly diagnosed glioblastoma with IDH-wildtype and unmethylated MGMT promoter status, confirmed by pre-RT molecular testing. Patients without a known MGMT status were excluded. All patients underwent at least one FU ^18^F-DOPA PET/CT scan. For each available PET FU scan, a corresponding conventional MRI set was also acquired, which has been used to assess progression with RANO criteria; however, not all MRI follow-ups had matched PET acquisitions. The absence of certain PET FU scans was attributable to factors such as ^18^F-DOPA synthesis failures, patient non-compliance with pre-scan protocols (e.g., fasting), or other logistical constraints. For analysis, patients were randomly assigned to training and testing cohorts in a 70:30 ratio. All FU images were organized by patient and separated into completely independent training and testing datasets accordingly. By this approach, no FU image of any patient in the test cohort was exposed in the training dataset.

### 2.2. Outcome Endpoints

This study aimed to assess the risk of tumor progression at each FU timepoint, recognizing that risk status evolves dynamically over time. FU PET images provide a temporal snapshot of the tumor’s state at each respective timepoint. However, conventional endpoints such as progression-free survival (PFS) or overall survival (OS) do not capture this time-varying nature of post-treatment tumor behavior. To address this, we introduced the concept of “remaining survival” (RS), defined as the time interval between a given FU imaging acquisition and the date of death, offering a more temporally resolved and clinically relevant outcome measure. Considering that median survival is reported to be 12 months or less from progression in the literature [11,26,32], FU scans were categorized as either “high-risk” or “low-risk” for progressive disease based on whether they had at least 12 months of remaining survival from the time of a given ^18^F-DOPA FU scan. A model-based risk transition timepoint was defined as the first FU timepoint when a “high risk” image was observed for a patient.

### 2.3. Radiomics Feature Extraction

PET scans were acquired using either a Discovery 690XT or Discovery MI PET/CT scanner from GE (Chicago, IL, USA), with a standardized acquisition protocol [16,35] to ensure consistency across patients. The reconstructed PET data were resampled into isotropic voxels of 2 × 2 × 2 mm prior to analysis. Feature extraction was performed with the PyRadiomics [36] platform (version 3.0.1), generating a comprehensive set of quantitative descriptors: 26 geometric (shape) features, 19 first-order statistical measures, and 70 texture metrics. The latter included 24 gray level co-occurrence matrix (GLCM), 16 gray level dependence matrix (GLDM), 16 gray level run length matrix (GLRLM), and 14 gray level size zone matrix (GLSZM) features, following the standardized definitions established by the Image Biomarker Standardization Initiative [37,38]. All shape descriptors were reported in voxel-based units. For reference, Appendix A summarizes the features extracted in this work.

Feature extraction was limited to regions of elevated tracer uptake, defined by a tumor-to-normal-brain (T/N) SUV ratio greater than 2.0. For the scan without T/N SUV ratio higher than 2.0, the features were automatically set to 0 by design, reflecting the absence of above-threshold uptake. This T/N ratio, established in a prior clinical trial [39] through biopsy correlation and multimodal image comparison, was prospectively adopted in our dose-escalation protocol to guide dose painting on gross tumor volume [16]. This high-uptake zone has previously been shown in our work to correspond to the biologically most aggressive portion of the lesion and to provide greater predictive value than the entire tumor volume [35]. To minimize inter-observer variability, tumor regions were delineated using an SUV-threshold-based autosegmentation approach. Each segmentation was subsequently reviewed by both an experienced medical physicist and a nuclear medicine physician, who carefully excluded areas of physiological uptake, such as striatal signals.

### 2.4. Radiomics Feature Selection

Feature selection is a crucial step in reducing input size to eliminate noise in data before machine learning training, and is particularly vital for small datasets. This study developed a multi-stage feature selection algorithm depicted in Figure 1. Exclusive to the training image set, the algorithm comprised three stages. Stage I evaluated the predictivity of each feature to remaining survival, categorizing feature values into two groups based on the optimal threshold, determined by the lowest *p*-value. Stage II involved 5 folds cross-validation of the features selected in Stage I to ensure their consistent predictivity across random subsets of the data. The underlying hypothesis was that a progressive or preventive feature should exhibit the same behavior across different subsets of the data. Both the entire training group and subsets consisting of FU1 images acquired at consistent timepoints after RT were considered in Stage II. Only features selected for all sets were deemed robustly predictive and advanced to Stage III. In Stage III, known as pair correlation filtering, the bivariate Pearson correlation coefficient was calculated for selected features categorized as shape, first order, and texture. If the correlation coefficient between two features exceeded 0.8, one of the features was eliminated. Priority was given to features that had been previously reported in the literature [11,26,32] to minimize the introduction of new features and leverage the additional robustness from independent testing using a similar, although not identical, amino acid PET tracer. Following these stages, only a limited number of features remained. While feature selection reduced noise, it also diminished information from discarded features. Further improvement on noise reduction and data orthogonality was achievable through feature compression and abstraction with manifold learning [36]. In this study, we utilized Uniform Manifold Approximation and Projection (UMAP) [37], a nonlinear dimension reduction technique to map the data structure in a high-dimensional space into a lower-dimensional space by preserving both global and local data structures. The selected features were projected into space of two features: UMAP x and UMAP y. As per standard practice in machine learning, a Standard Scaler, implemented in Scikit-Learn [40] (version 1.5.1), was applied to the projected features before feeding into the ML models.

In order to assess the efficacy of the feature selection pipeline and the individual contributions of its components, ablation tests were conducted by systematically removing elements of the pipeline. Ablation Test 1 involved the elimination of the third stage of feature selection, resulting in a larger set of selected features prior to manifold learning. This test aimed to evaluate the ability of manifold learning to manage higher-dimensional data. Ablation Test 2, on the other hand, excluded manifold learning before machine learning to assess model performance in the absence of feature abstraction.

### 2.5. ML Algorithms

Two machine learning (ML) approaches were employed to explore the relationship between the features selected from the FU ^18^F-DOPA PET images and remaining survival. The first approach used AutoML with Amazon’s AutoGluon [41] (version 1.0.0), a new open-source framework of machine learning models. Auto ML is designed to automatically select the appropriate ML algorithm and optimize hyperparameters. The second ML algorithm employed was the Explainable Boosting Machine (EBM) which offers the advantage of producing interpretable model outputs. The model was built using Microsoft (Seattle, WA, USA)’s InterpretML library [42] (version 0.5.0). This is an implementation of the Generalized Adaptative Model (GA^2^M) [43] which replaces single variables in a linear model with single-feature models (shape functions) that utilize bagging and gradient boosting, as well as additional terms for feature pairing interactions. Because of its additive nature, the contribution of each feature to the model and to the classification for each individual patient can be easily quantified and visualized. It has been shown to have comparable classification performance [42] with other popular models like Random Forest and XGBoost, but with much better interpretability.

Both models examined in this study underwent validation using an independent test dataset, which comprised FU PET scans from the patients whose information was excluded from the model training process.

For comparison, we implemented a univariate baseline using TBR_max_ (maximum T/N SUV ratio per scan), a reported predictive marker in multiple amino-acid PET glioma studies [32,44,45,46]. The decision threshold was selected on the training set (via a simple grid search to maximize accuracy) and then fixed and applied to the independent test set for evaluation.

## 3. Results

### 3.1. Patient Cohort and Servalliance Images

A total of 33 patients were identified with the inclusion criteria stated. The span of OS ranged from 5 months to 41 months, with a median at 15 months. Ages of patients ranged from 19 to 77 years with a median of 55 years, comprising 58% male and 42% female. The number of ^18^F-DOPA FU images for a given patient ranged from 1 to 14 with a median of 3. The first FU images were acquired consistently at 1 month after completing RT (ranging from 27 days to 42 days with a median of 33 days). Subsequent ^18^F-DOPA FU frequency determined for each patient based on the latest available FU images. The training and testing sets have 103 FU ^18^F-DOPA images from 25 patients and 37 FU ^18^F-DOPA images from 10 patients, respectively. The patients in the testing set were not exposed to the training set.

### 3.2. Radiomics Feature Selection

Starting with 115 radiomics features extracted from PET images, individual predictivity evaluation reduced the pool of the features to 104. After cross-validation stages, a total of 47 features (including 11 shape features, 12 first-order features, and 24 texture features) were identified. These features demonstrated the potential to stratify patients into distinct groups with statistical differences, following the application of an optimized threshold. After the third stage of pair correlation filtering, only 6 features, including 3 shape features, 1 first order feature and 2 texture features, were identified. They are summarized in Table 1 and their brief explanation can be found in Appendix A. The dimension reduction stage with UMAP further reduces those 6 features into 2 projected features (UMAP x and UMAP y) and their visualization is plotted in Figure 2. The feature count reduction flow is also summarized in Appendix A.

### 3.3. Model Prediction

Utilizing the two projected UMAP features, Autogluon (version 0.5.0) was used to train 219 classification models with high-quality presets, encompassing Random Forest, Light GBM, and various ensemble models, among others, based on the training dataset. The evaluation metric used was the Area Under Curve (AUC), and the best-trained model (WeightedEnsemble_L3) achieved an AUC of 0.81 when applied to the independent test data. Based on 2000 bootstrap resamples, the model’s AUC had a 95% confidence interval of [0.650, 0.942]. WeightedEnsemble_L3 is a three-level ensemble model employing Autogluon’s multi-layer stacking strategy [41]. At the bottom layer, it consists of 110 individual models with nonzero weights, which feed into the second layer composed of 7 ensembled models, ultimately contributing to the top layer (WeightedEnsemble_L3). A schematic of the WeightedEnsemble_L3 architecture and its receiver operating characteristic (ROC) curve on the test dataset are depicted in Figure 3a and Figure 3b, respectively.

The EBM model was also trained with the same training dataset. When it was applied to the test data, it achieved the AUC of 0.83. Similarly, based on 2000 bootstrap resamples, the EBM model’s AUC had a 95% confidence interval of [0.681, 0.944]. The ROC curve on the test dataset based on EBM model is shown in Figure 3b.

A sensitivity test to the trained models was also performed by redefining high risk status to RS ≤ 10 months and RS ≤ 14 months. The AUCs of AutoGluon are 0.80 and 0.81, respectively. The AUCs of EBM are 0.82 and 0.83, respectively.

On the training set, the baseline model identified an optimal TBR_max_ threshold of 2.16, yielding an AUC of 0.72. Applied unchanged to the independent test set, it achieved an AUC of 0.66.

Ablation Test 1, which omitted Stage III feature selection, and Ablation Test 2, which excluded manifold learning, both resulted in inferior model performance when assessed using the reserved test dataset. In Ablation Test 1, Autogluon achieved an AUC of 0.71, while EBM attained 0.68. Conversely, in Ablation Test 2, Autogluon achieved an AUC of 0.80, while EBM reached 0.73.

To depict the changes in tumor biology associated with the RS of each patient, we proposed a RS map, which represents the temporal evolution of a patient’s remaining life with respect to the pre-RT imaging timepoint. Figure 4 showcases the predictions of RS risk for each PET image in the test dataset, utilizing the EBM model. In this figure, the pre-RT PET image is designated as timepoint 0 for each patient. Each square on the RS map corresponds to a subsequent ^18^F-DOPA PET FU imaging timepoint. PET scans that are predicted to be high-risk according to the model are depicted as dark blue squares, while low-risk PET scans are represented by light blue squares. For reference, a cyan diamond is provided to indicate the timepoint at which radiographic progression was determined solely based on the RANO criteria derived from conventional MRI. As shown in the figure, the model is also applied to pre-RT images, which are not included in the training dataset. Although pre-RT images are very different from FU images because they are followed with interventions, it is reasonable to assume that pre-RT images should present high risk for survival if intervention (multimodal therapies) was not applied. The model prediction is consistent with this assumption, classifying 86% of pre-RT images as high risk.

The EBM model’s predicted transition timepoints from low-risk to high-risk status were compared with the radiographical progression timepoints determined using RANO criteria based on conventional MRIs for the entire cohort, as shown in Table 2. The MRIs were typically obtained within a day before or after the ^18^F-DOPA scans to ensure simultaneous assessment. Based on RANO progression calls, the mean RS of the cohort from the timepoint of the progression calls was found to be 8.0 months, with a standard deviation of 6.2 months. In contrast, based on the model predictions, the mean RS at the timepoint of model-predicted risk transition was 10.0 months, with a standard deviation of 5.6 months, excluding the patients without a model-predicted risk transition. Among the patients in this study, 33.3% had RANO progression calls coincident with model-predicted risk transition timepoints. This subgroup had an average survival of 10 months with a standard deviation of 6.2 months after the RANO progression timepoints. 91% of this coincident subgroup had remaining survival less than or equal to 12 months after the RANO progression call. For 45.4% of patients, the model predicted risk transition occurred earlier than the RANO call. 6.1% of patients had a model predicted risk transition timepoint later than the RANO call. In the remaining 15.2% of patients, no ^18^F-DOPA scans were acquired after the RANO call, and all ^18^F-DOPA scans prior to and coincidence with the RANO call timepoint were predicted to be low risk, so a risk transition timepoint was not reached for the available scans. Only 40% of this subgroup had RS ≤ 12 months after the RANO progression call. Averaged remaining survival and the percentage of RS ≤ 12 months are reported in Table 2.

Both AutoGluon and EBM provide feature-importance estimates to aid interpretability. Applied to the independent test set, the importances of the UMAP-projected features are summarized in Table 3; for comparability, values are normalized to sum to 1. Because AutoGluon does not include explicit pairwise interaction terms, the “interaction” entry is zero for AutoGluon, whereas EBM reports contributions from both main effects and modeled pairwise interactions.

The EBM model also provides a means to evaluate the contribution of each feature to the model prediction at an individual level. Figure 5 illustrates an example of feature interpretation for a patient in the test cohort (RT_FDOPA40). The probability of being classified as high risk is calculated for each image, and the contributions (weights) of the shape function for each feature or the interaction between two variables are plotted quantitatively. A positive value indicates favoring a high-risk classification, while a negative value favors a low-risk classification. The final decision is a balance between all of the features.

To further assess clinical utility, we conducted a decision curve analysis (DCA). The results are presented in Figure 6a. The Kaplan–Meier survival curves comparing progression-free survival (PFS; RANO-defined) with model-predicted low-risk survival (MLRS) are presented in Figure 6b.

## 4. Discussion

Differentiating true tumor progression from treatment-related effects remains a persistent and unresolved challenge in the post-treatment surveillance of glioblastoma. The RANO criteria, which rely on conventional MRI sequences, have limited accuracy—particularly in the context of escalated radiation doses. In the clinical trial involving dose escalation, PFS based on RANO criteria has shown no correlation with OS, suggesting that many progression assessments may be confounded by treatment effects. These limitations underscore the need for more informative imaging modalities to improve the accurate identification of true progression. In this study, we evaluated the utility of longitudinal post-treatment ^18^F-DOPA PET imaging for predicting RS in glioblastoma patients. Given the reported correlation between pathologically confirmed PFS and OS [26,32], We explored the use of RS as a surrogate but dynamic marker for OS to capture the evolving risk status of the tumor. Our findings indicate that incorporating ^18^F-DOPA PET into post-treatment surveillance can enhance clinical decision-making by providing additional prognostic value. This PET-based model has potential clinical value in prognostication and may support more personalized decisions regarding salvage treatments.

In outcome analysis for relatively small datasets, for example, for uncommon cancers like glioblastoma, feature selection and dimensionality reduction are essential for mitigating noise and uncovering meaningful patterns in high-dimensional data. While selecting only the most predictive features can reduce noise, it risks discarding complementary or interacting information. Conversely, applying dimensionality reduction directly to large, noisy feature sets can obscure relevant signals. Striking a balance between noise reduction and information preservation is a core challenge in machine learning and an area of active research. To address this, we developed a tailored pipeline that first identifies robust predictive features to narrow the input space, followed by manifold learning using UMAP for further dimensionality reduction and feature abstraction. This combination preserved key information while minimizing redundancy, leading to optimal performance across models, as shown in Figure 3. Ablation tests demonstrated that injecting too many raw features prior to manifold learning (Ablation Test 1), or omitting manifold learning altogether (Ablation Test 2), consistently degraded model performance in both AutoML and EBM frameworks. Beyond improving model accuracy, UMAP also enabled visualization of the high-dimensional feature space (Figure 2), facilitating the identification of feature patterns and clusters that would otherwise be hidden in high dimensional data. Notably, patients with shorter remaining survival (RS) tended to cluster in regions with lower UMAP x and y values, while those with longer RS exhibited the opposite trend, suggesting distinct underlying feature distributions. Although manifold learning techniques like UMAP have been widely applied in genomics (for example, in clustering tumor subtypes), they remain underutilized in radiomics, particularly with diagnostic imaging data. To our knowledge, this study represents the first application of UMAP to PET radiomics in glioblastoma, demonstrating both its analytical and interpretive value.

Both the AutoML (Autogluon) and EBM models demonstrated reasonable and comparable ROC_AUC. The AutoML approach takes care of many machine learning details in the background, including data scaling, model selection and ensemble, as well as cross validation. It significantly reduces the domain knowledge related to machine learning and enhances the utilization of machine learning tools. The multiple layer architecture design of Autogluon, with model ensembles, also helps to maintain the model’s robustness. The performance degradation in the ablation tests is less severe compared to that of the EBM model. For comparison, a univariate baseline model using TBR_max_, a metric reported as prognostic in multiple PET glioma studies [32,44,45,46], performed weaker than both AutoML and EBM on the training and test sets, though it remained predictive; notably, the optimal TBR_max_ threshold selected on the training set was similar to the optimal cutoffs reported elsewhere [32,45]. The superior performance of AutoML and EBM underscores the added value of multivariable modeling over single-parameter approaches. Furthermore, the AUCs achieved with AutoML and EBM are comparable to, or modestly higher than, previously reported multivariable radiomics model [32]; however, direct comparison is not appropriate given differences in PET tracers, cohort composition, and treatment protocols. Because our cohort underwent ^18^F-DOPA-guided dose escalation, evaluation and external validation in patients treated with standard-dose regimens are warranted as such data become available.

Both AutoGluon and EBM provide native feature-importance estimates (Table 3). The resulting profiles differ by algorithm: AutoGluon places most of the weight on UMAP-x and does not model interaction terms, whereas EBM, while also ranking UMAP-x highest, shows a more balanced distribution across features and includes an explicit UMAP x–y interaction effect. The EBM model, known for its interpretability, also provides valuable insights into how each feature contributes to the model’s decision for individual images. This information is particularly useful for clinicians in evaluating changes in tumor status, as it transforms the machine learning model from a black box into a glass box, as illustrated by the example in Figure 5. This interpretability provides clinicians with a deeper understanding of the model’s decision-making process and highlights the evolution of critical predictive features. By offering more detailed information beyond a simple “yes” or “no” classification, this interpretability empowers clinicians in making informed clinical decisions based on the model’s insights.

In the decision-curve analysis on the independent test set (Figure 6a), both AutoGluon and EBM showed positive net benefit across clinically plausible thresholds (~0.10–0.40), outperforming “treat-all” and “treat-none.” AutoGluon provided greater benefit at lower–mid thresholds (~0.10–0.35), consistent with early-warning use, whereas EBM was comparable and modestly higher at higher thresholds (~0.40–0.50). These findings suggest that model-guided surveillance offers greater clinical utility than default strategies across common operating points, with the threshold tailored to local risk tolerance and practice patterns. Complementing this, Kaplan–Meier curves (Figure 6b) comparing RANO-defined PFS with model-predicted low-risk survival (MLRS) show distinct temporal patterns consistent with earlier model signaling in a subset of patients. Taken together, the DCA and KM results support the potential for actionable early warnings, warranting confirmation in larger, multi-institutional cohorts.

Based on ^18^F-DOPA PET images, the RS map suggested in this work provides a tool to monitor the patients’ status associated with their survival, as a surrogate to tumor progression. With significant treatment complications and high rate of incorrect progressional calls, especially for patients treated with escalated radiation dose, the remaining survival becomes a more reliable indicator than the PFS determined by the conventional methods. ^18^F-DOPA PET images can provide a superior sensitivity to detect tumor progression. For the patients included in this study, 41.2% showed earlier indications from ^18^F-DOPA PET than the conventional MRIs, and their shorter survival after progressional call also confirmed the better sensitivity of ^18^F-DOPA PET. In clinical practice, earlier detection of progression risk transition could alter management in practice: prompting earlier confirmatory imaging and consideration of salvage therapies (e.g., re-resection, re-irradiation, systemic therapy, or trial enrollment). For the patients with later indication from ^18^F-DOPA PET, their survival after progressional call is significantly longer than the average but their RS after model predicted risk transition is consistent with the average RS, which suggests that ^18^F-DOPA PET may also have better capability to distinguish true tumor progression from treatment effects.

In this study, we introduce RS as a pragmatic endpoint and use risk transition with respect to RS as a surrogate marker of progression—information that is readily obtainable and clinically manageable when histopathologic confirmation is not feasible. Guided by prior reports, we adopted RS ≤ 12 months as the primary threshold; sensitivity analyses across 10–14 months yielded consistent findings, indicating robustness to threshold selection. We emphasize that RS is not a substitute for formal progression adjudication (e.g., RANO) but provides complementary, time-anchored evidence when progression determination is challenging. Future work is necessary to prospectively validate the relationship between RS-based risk transitions and pathology-confirmed progression.

Although our work has shown promising applications of ^18^F-DOPA PET images and the radiomics modeling based on ^18^F-DOPA PET, this study has several limitations that should be acknowledged. Firstly, the small sample size poses a primary limitation. A larger sample size, ideally from cross-institutional data, will be extremely helpful to validate the ML models and improve their accuracy for personalized medicine. As an amino-acid PET tracer that is not FDA approved, the available data are still very scarce and will take time to accumulate more data, but we optimistically expect that more evidence of the strength of ^18^F-DOPA PET will continue to be demonstrated. Secondly, our investigation only included patients treated with escalated RT dose, and it remains unclear whether the models and prognostic delta features would hold consistent for a patient cohort treated with a standard dose protocol. Although we believe that ^18^F-DOPA PET imaging can be a promising tool for the surveillance of glioblastoma treated with standard dose protocol, prospective trials are required to determine its clinical impact and integration pathways. Thirdly, our study focused exclusively on patients with unmethylated MGMT and wild-type IDH1, reflecting where adequate, consistent longitudinal ^18^F-DOPA PET FU imaging was available. In accordance with WHO CNS5 (2021) excluding IDH1-mutant glioma from glioblastoma, our cohort focuses on IDH1-wildtype. As noted in prior studies [16,47,48], patients with IDH mutation or MGMT promoter methylation often have longer survival trajectories; however, in our dose-escalation cohort, ^18^F-DOPA PET FU imaging in these subgroups was frequently incomplete or discontinued after RANO-based progression determinations, yielding too few complete longitudinal series for reliable stratified modeling. Therefore, in this study, we have not explored the differences in radiomic features between cohorts with different histologic biomarkers, which could potentially provide valuable insights. Fourthly, although the manifold learning combined with feature selection turns out to be an effective way to reduce noise and makes high dimensional data visible, it comes with a price too. The manifold learning decreases the interpretation of the features and makes it less straightforward to understand what changes in an image contributes to the model decision. Of course, there is no guarantee that such a dominant and interpretable feature may exist at all. Noise reduction, feature abstraction and interpretability are all very active research topics in machine learning, and they hold very critical meanings to outcome studies, including our work. Given these limitations, this work should be regarded as exploratory and hypothesis-generating, and it warrants subsequent studies in larger, more heterogeneous cohorts and across diverse radiotherapy protocols to evaluate generalizability.

## 5. Conclusions

Radiomic features extracted from PET images acquired with the amino acid tracer ^18^F-DOPA offer a unique tool to capture glioblastoma tumor status after radiotherapy treatment and can be used to monitor tumor progression and potentially improve upon standard MRI surveillance. Based on radiomics features extracted from automatically contoured tumor core volumes with a high T/N SUV ratio, this work focused on building a radiomics model to predict RS, which is used as a practical surrogate for tumor progression.

With an in-house developed feature selection algorithm followed by manifold learning, the prediction model for RS was constructed using two machine learning algorithms: Auto-ML and Explainable Boost Machine. These models demonstrated promising performance, achieving AUC of 81% and 83%, respectively, when evaluated on an independent test set. Notably, the models also exhibited high accuracy (>80%) in identifying aggressive tumor biology when applied to pre-RT images, which were not included in the training set. These findings suggest that the radiomics features extracted from FU ^18^F-DOPA PET scans can serve as valuable additions to the traditional conventional MR approach, which lacked correlation between overall survival (OS) and progression-free survival (PFS) in detecting tumor progression. Moreover, the proposed remaining survival map provides a tool for monitoring the evolution of tumor progression.

Moving forward, our research aims to expand in two directions. Firstly, we plan to investigate the MRI data, specifically diffusion and perfusion MRI sequences acquired concurrently with PET images. This analysis will provide further insights into tumor heterogeneity and its temporal changes, complementing the findings from this study. Secondly, we aim to broaden the cohort size by incorporating data from different protocols and institutions. As part of this effort, we are organizing a cross-institutional study to utilize ^18^F-DOPA PET, with planned harmonization of acquisition/reconstruction parameters, centralized data curation/QA, and a prespecified analysis pipeline. This collaboration will prospectively enroll patients for ^18^F-DOPA-guided dose escalation radiotherapy, enabling external validation and assessment of generalizability across centers. This expansion will enable us to validate the robustness of our models and prognostic features and facilitate the development of multivariable models for more personalized medicine approaches. We firmly believe that amino-acid PET tracers, including ^18^F-DOPA, hold immense potential to enhance glioblastoma treatment and management, presenting exciting research and clinical opportunities.

## Figures and Tables

**Figure 1 cancers-17-03560-f001:**
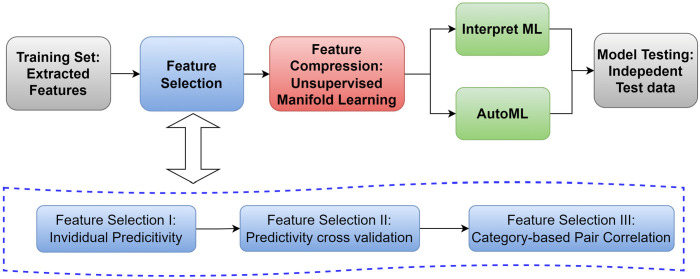
High level schematic workflow for feature extraction, selection, compression, model training and testing.

**Figure 2 cancers-17-03560-f002:**
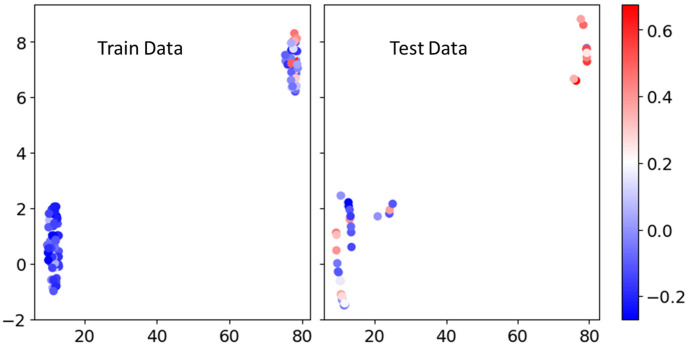
UMAP Visualization of Selected Features. The figure displays a visualization of the 2 projected UMAP features derived from the 6 features selected after pair correlation filtering. Each data point is colored to indicate the deviation of the RS from 12 months. RS less than 12 months (high risk) is represented in blue, with darker blue indicating shorter remaining survival. Conversely, RS longer than 12 months (low risk) is represented in red, with deeper shades of red suggesting longer remaining survival.

**Figure 3 cancers-17-03560-f003:**
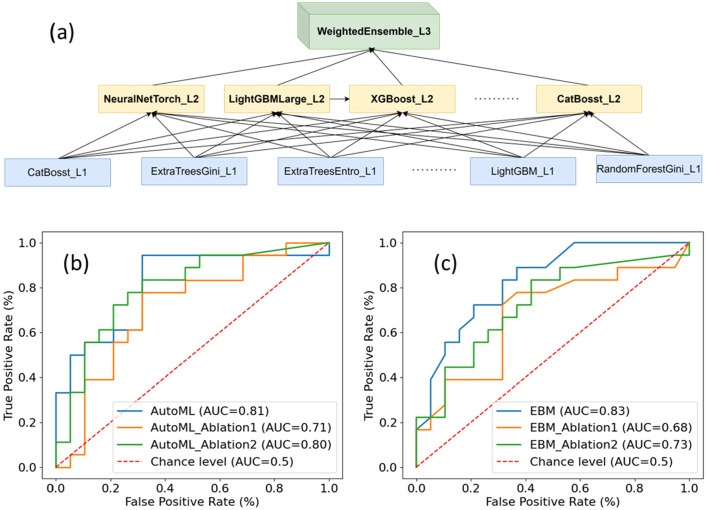
ML architecture and performance. (**a**) Multiple layer architecture of Autogluon WeightedEnsemble_L3 model. The bottom layer has 110 models. The second layer has 7 models. (**b**) AUC_ROC curve of the test data for Autoglueon ML model. Ablation 1 is the test without Stage III of the feature selection in the pipeline; Ablation 2 is the test without manifold learning after feature selection. Chance level is random guess. (**c**) AUC_ROC curves of the test data for the EBM model.

**Figure 4 cancers-17-03560-f004:**
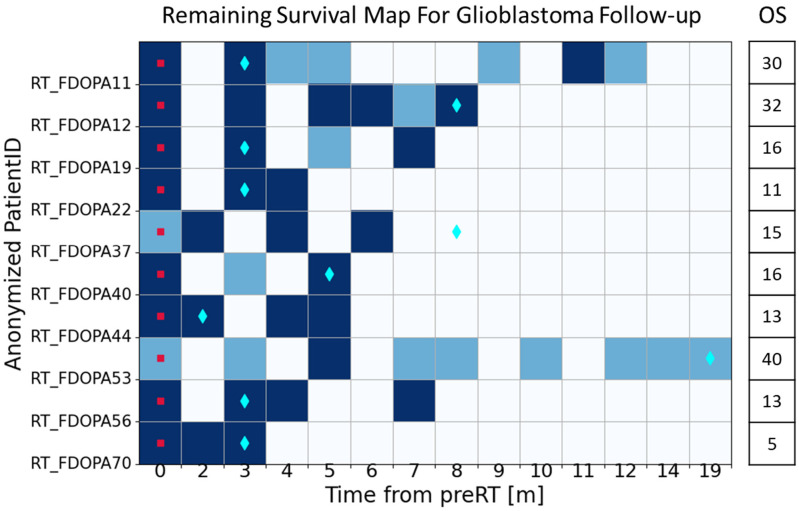
EBM model predicted FU map for each patient from the test cohort based on their pre-RT and FU PET images. The X axis reflects the elapsed time in months from pre-RT imaging to the respective FU imaging timepoint. Y axis is the anonymized patient ID. For each Pre-RT or FU PET image timepoint, dark blue shading indicates a remaining survival prediction by the model to be high risk, whereas light blue shading indicates a low-risk prediction. Cyan diamonds represent the RANO radiographic progression timepoint as determined by conventional MR images. Red square represents the pre-RT images. OS for each patient is listed in the far-right column.

**Figure 5 cancers-17-03560-f005:**
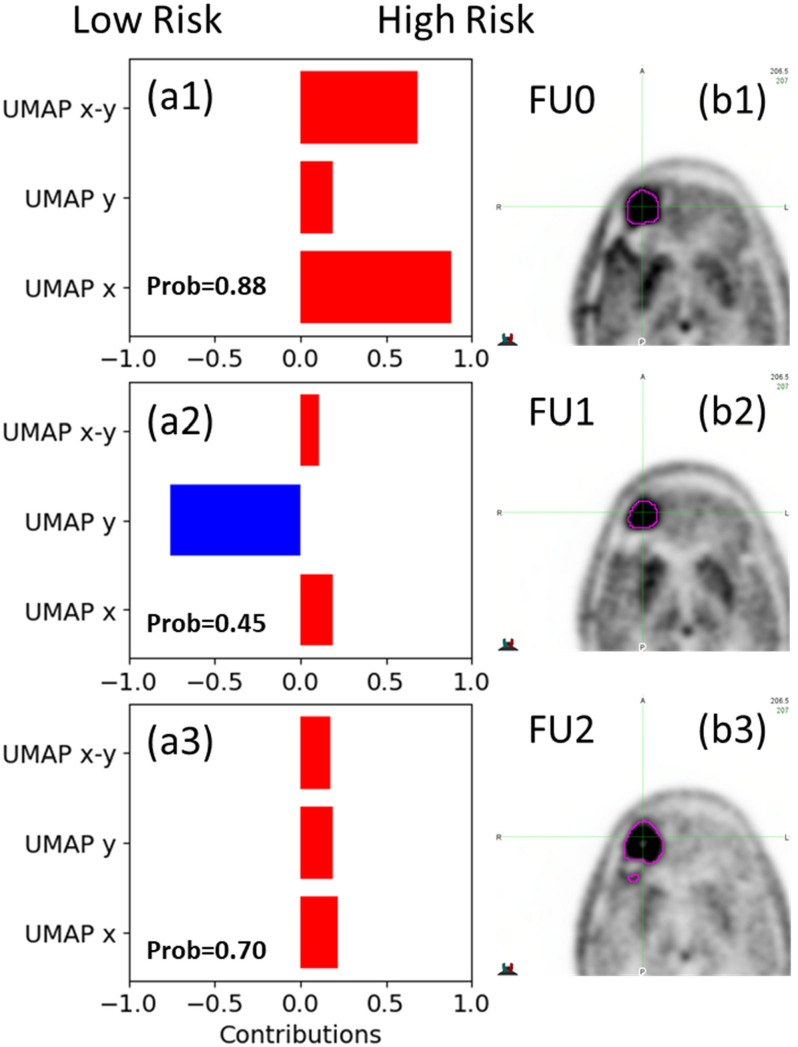
The contribution of each feature or feature interaction to the model decision for (**1**) Pre-RT (FU0), (**2**) FU1 and (**3**) FU2 images of test patient RT_FDOPA40. The left column (**a**) shows the weights of each shape function, including both single variable and two-variable interactions, at each image acquisition timepoint. Values above 0 favor a high-risk classification and values below 0 favor a low-risk classification. The right column (**b**) illustrates a single slice of the tumor at the corresponding timepoints.

**Figure 6 cancers-17-03560-f006:**
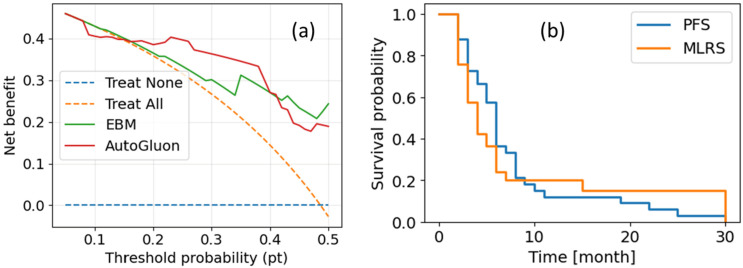
Clinical utility: (**a**) Decision curve analysis on the independent test set. Net benefit (y axis) is plotted against threshold probability (x axis) for AutoGluon (red) and EBM (green), with treat-all (orange dashed) and treat-none (blue dashed) reference strategies. Both models yield positive net benefit across clinically plausible thresholds. (**b**) Kaplan–Meier curves comparing progression-free survival (PFS; RANO-defined) with model-predicted low-risk survival (MLRS).

**Table 1 cancers-17-03560-t001:** Selected Features with Optimized Cut Points and *p*-values. Selected features identified using the in-house developed feature selection algorithm, along with their corresponding optimized cut points for categorizing high vs. low risk, and associated *p*-values between low risk and high risk data. The feature selection process is entirely based on the training dataset.

Category	Features	Optimized Cut Point	*p*-Value
Shape	MeshVolume	10.114	<0.001
	MajorAxisLength	0.477	<0.001
	LeastAxisLength	0.194	<0.001
First Order	Variance	0.001	<0.001
Texture	[GLCM] Correlation	0.003	<0.001
	[GLRLM] LongRunEmphasis	0.499	<0.001

**Table 2 cancers-17-03560-t002:** Comparison between the EBM model predicted high-risk to low-risk transition timepoint based on ^18^F-DOPA PET images and the RANO progression call based on conventional MR images for different subgroups. For each category, the percentages of patients whose remaining survival is equal or less than 12 months after RANO progression calls or model predicted risk transition are also given.

	MP Coincident with RANO	MP Earlier than RANO	MP Later than RANO	Missing Transition
Percentage of the cohort population	33.3%	45.4%	6.1%	15.2%
RS after RANO PC (months)	10.4 ± 6.2	7.1 ± 5.3	16.5 ± 3.5	14.3 ± 5.7
% (RS ≤ 12 months) after RANO PC	91%	87%	0%	40%
RS after MP Transition (months)	10.4 ± 6.2	10.1 ± 5.9	10.5 ± 0.7	N/A
% (RS ≤ 12) after MP Transition	91%	80%	100%	N/A

MP: Model Prediction. PC: Progressional call. RS: Remaining survival.

**Table 3 cancers-17-03560-t003:** Feature-importance comparison for the AutoGluon AutoML and EBM models. Importances are computed using each model’s native estimator and normalized to sum to 1. The pairwise interaction term (UMAP x–y) is reported only for EBM and is not available in AutoGluon.

Features	Autogluon	EBM
UMAP x	0.910	0.417
UMAP y	0.07	0.292
UMAP x–y	0	0.291

## Data Availability

Data generated and analyzed during this study are included in this article. Extracted radiomics features and the source code are available upon request.

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
