# Peer review of "Predicting Remaining Survival of Glioblastoma Patients with Radiomics Analysis Based on ^18^F-DOPA PET Images"

_cancers, 2025, doi:10.3390/cancers17213560_

Round 1

Reviewer 1 Report

Comments and Suggestions for Authors

1. The number of cohort (35 patients, 140 total PET scans) is very small, which restricts the potential of the study to be statistically powerful and generalizable.
2. Internal validation only (70:30 train/test split) was carried out.
3. The authors might comment on the intention to perform multi-center validation and restrictions on the relevance to standard radiation protocols or patients having other molecular biomarkers.
4. It is recommended to add more detail as to the nature of the correlation between certain radiomic features (e.g., MeshVolume, GLCM correlation) and tumor biology or progression, albeit in a simplified form.
5. Add quantitative sensitivity and specificity or time to detection improvement over RANO, perhaps in a table.
6. To check the robustness, a sensitivity analysis of SUV threshold selection (T/N>2.0) may be considered.
7. Discuss further more the limitations because this excludes patients with varying IDH/MGMT status.
8. Although reported, the manuscript does not directly compare ROC AUC with other PET tracers, or conventional imaging models, which could inhibit perceived effect.
9. Talk about the performances of models in standard RT or mixed patient groups.
10. Mark which characteristics make predictions and discuss SHAP or other explainability techniques, along with EBM.
11. The quality of figure 1 needs to be enhanced.
12.  compare to baseline models.

Author Response

We express our heartfelt gratitude to both reviewers and the editors for their invaluable comments, which have significantly contributed to enhancing the quality of this manuscript. We also sincerely appreciate the editor’s kind consideration in granting us additional time, which allowed us to thoroughly address each of the reviewers’ suggestions in this revised submission. The manuscript has undergone thorough revisions in accordance with the provided feedback, and the responses to the reviewers' comments are summarized below. Within this document, responses are marked in blue for easy reference, and corresponding changes in the manuscript are highlighted in red and annotated with the respective reviewer comment identifier. For instance, "R1-C2" denotes the revision made in response to the second comment (C2) from Reviewer 1 (R1).

  1. The number of cohort (35 patients, 140 total PET scans) is very small, which restricts the potential of the study to be statistically powerful and generalizable.
    Response (R1-C1): We sincerely thank the reviewer for pointing out this limitation. We acknowledge that the small sample size is a key limitation of our study, and we address this in the forefront of the discussion section. In the revision, we have (i) explicitly framed our analyses as exploratory/hypothesis-generating, and (ii) tempered the Abstract (Conclusions) and Discussion language to avoid over-interpretation.
    There are practical reasons for the cohort size that we now state more clearly. Glioblastoma is a relatively uncommon tumor, which often results in small sample sizes, particularly when exploring novel imaging techniques. 18F-DOPA is not yet FDA-approved for glioblastoma; therefore, access is constrained by research funding and institutional resources. In addition, to our knowledge this is a unique clinical dose-escalation trial guided by 18F-DOPA PET, making accrual intrinsically challenging. We have clarified these points in the limitation paragraph of Discussion.
    Methodologically, we took steps to mitigate overfitting given the sample size: we restricted model complexity through a multi-stage approach (initial univariable screening of radiomic features with cross validation followed by manifold learning to reduce the feature space and degrees of freedom before modeling). This strategy is intended to improve robustness in small-sample settings.
    Finally, we have added text outlining our ongoing plans to enable broader validation: a new prospective study with additional patients for 18F-DOPA PET guided radiotherapy is underway, and we are actively pursuing multiinstitutional collaboration to expand the cohort and perform external validation. We have also included contextual references noting that several prior radiomics studies in glioblastoma (e.g., Ref. 32, n=34) were necessarily small yet still generated clinically informative hypotheses; we cite these to contextualize feasibility rather than to justify generalization. We hope these clarifications and revisions address the reviewer’s concern and transparently convey the scope and limitations of our current findings.
  2. Internal validation only (70:30 train/test split) was carried out.
    Response (R1-C2): We thank the reviewer for this valuable observation and agree that reliance on internal validation is a limitation. In the revision, we explicitly state this up front in the discussion section and emphasize that our findings are exploratory/hypothesis-generating and not yet generalizable without external validation.
    To improve transparency, we detail the validation protocol in our method section: the split was performed at the patient level (not scan level) to prevent information leakage across the 140 longitudinal PET scans; the held-out 30% test set was not used for any model selection or tuning; and all preprocessing, feature
    screening, and manifold learning steps were fit exclusively on the training data prior to a single and final evaluation on the test set. This design was chosen to avoid cross-patient leakage given repeated measures and to preserve an untouched set for performance estimation in a small cohort.
    As noted in R1-C3 and now stated in the manuscript, we are actively pursuing multiinstitutional collaboration and have initiated a prospective study to accrue additional patients, with the explicit goal of conducting external, multi-center validation of the proposed approach. We hope these clarifications and our concrete plan address the reviewer’s concern.
  3. The authors might comment on the intention to perform multi-center validation and restrictions on the relevance to standard radiation protocols or patients having other molecular biomarkers.
    Response (R1-C3): We thank the reviewer for raising this important point. In the revision, we expand the discussion section to address (i) our plans for multi-center validation and (ii) the relevance of our approach to standard radiation protocols and to patients with differing molecular biomarkers.
    We agree that external validation across institutions is essential for proof of generalizability. We are in active discussions with the University of Michigan to develop a multi-center study of 18F-DOPA-guided radiotherapy; a proposal for a prospective national trial is also under review by NRG Oncology. We are aligning research aims, funding, data-sharing governance, and cross-institutional quality assurance of workflows and data. Our goal is to generate a larger, more comprehensive dataset to test the hypotheses advanced here and provide robust evidence for the clinical utility of 18F-DOPA PET imaging for glioblastoma. These plans are now summarized in the Discussion.
    Our dose-escalation framework was implemented on a standard-of-care backbone (conventional fractionation with concurrent/adjuvant temozolomide) with selective dose escalation to 18F-DOPA–avid subregions, representing a protocol deviation confined to the escalation component. We clarify in the revision that the proposed 18F-DOPA PET surveillance workflow is designed to function as a complementary decision-support tool and can be applied to patients treated with standard protocols; however, prospective trials are required to determine its clinical impact and integration pathways. These clarifications are now included in the Conclusion section.
    Because 18F-DOPA is not FDA-approved for glioblastoma, access and reimbursement are limited, which constrains follow-up frequency and can bias accrual toward patients with poorer-prognosis profiles and short survival. In the planned multi-center effort, we will prospectively collect comprehensive molecular data (e.g., IDH, MGMT promoter methylation) and address the stratification beyond various baseline molecular markers. In the revised manuscript, we now articulate these considerations as context (not justification) and position our work as hypothesis-generating toward a more definitive, multi-center validation. We hope these additions address the reviewer’s request and clarify the intended clinical scope of the proposed approach.
  4. It is recommended to add more detail as to the nature of the correlation between certain radiomic features (e.g., MeshVolume, GLCM correlation) and tumor biology or progression, albeit in a simplified form.
    Response (R1-C4): We sincerely appreciate the reviewer’s suggestion to improve the clarity of the presentation of our work. Radiomics features are typically defined mathematically to capture aspects of shape, intensity, or texture, and some may not be immediately intuitive. In the revised manuscript, we have added two supplementary tables in the newly provided Supplementary Materials: Supplementary Table S1, which lists all features extracted in this study, and Supplementary Table S2, which provides brief, plain-language descriptions of selected features (including MeshVolume and GLCM correlation) to aid interpretation. We did not incorporate these descriptions directly into Table 1 to keep the main table concise
    and readable in the primary text. A hyperlink to more detailed and mathematical descriptions of these features is also included in the Supplementary Materials.
  5. Add quantitative sensitivity and specificity or time to detection improvement over RANO, perhaps in a table.
    Response (R1-C5): We thank the reviewer for this constructive suggestion. We agree that quantitative comparisons are valuable; however, classical sensitivity/specificity requires a well-defined ground truth (progression vs. no progression) at each imaging timepoint. In glioblastoma, pathological confirmation is rarely feasible for serial surveillance due to invasiveness and patient condition, and once RANO progression is declared, subsequent scans are typically not adjudicated against RANO, leaving “negative” timepoints ambiguously defined. This limits the validity of conventional 2×2 analyses.
    We agree with the reviewer that time-to-detection improvement directly reflects a clinically actionable goal (identifying progression risk earlier to potentially enable earlier intervention) and is therefore a key metric that our study is specifically designed to address. We focused on timeliness of detection and a survival-based surrogate. Specifically, for patients with adequate follow-up, we used 12-month residual survival after the detection event as a proxy for biological progression. In terms of timing, 45.4% of patients were identified as having a model prediction transition earlier than RANO, with mean remaining survival from detection increasing from 7.1 months (RANO) to 10.1 months (model), consistent with earlier detection. Conversely, 6.1% were identified as having a model prediction transition later than RANO, with mean remaining survival of 10.5 months (model) versus 16.5 months (RANO) in that subset, which may indicate fewer false-positive early progression calls. Given sample size and the surrogate nature of the endpoint, we avoided strong inferences.
    In the revision, we have expanded Table 2 to report the proportion of patients with residual survival ≤12 months for each method and to summarize earlier/later detection percentages. We respectfully submit that, under the constraints of imperfect ground truth in longitudinal glioblastoma surveillance, these surrogate-based and timing-oriented metrics offer a meaningful and clinically relevant way to compare our model with RANO, while transparently acknowledging their exploratory nature.
  6. To check the robustness, a sensitivity analysis of SUV threshold selection (T/N>2.0) may be considered.
    Response (R1-C6): We sincerely appreciate the reviewer’s thoughtful suggestion. In this study, the T/N > 2.0 SUV threshold was pre-specified a priori based on prior prospective, biopsy-correlated work (Ref. 39) and subsequently supported by another study (Ref. 35) demonstrating superior sensitivity/predictive performance relative to lower cutoffs. Importantly, this threshold served as the clinical definition for delineating the Gross Tumor Volume (GTV) that received dose escalation in our trial; as such, it has direct therapeutic meaning rather than functioning as a tunable modeling parameter.
    Within this context, varying the threshold post hoc would effectively redefine the biological target volume and decouple our analyses from the actually treated subregion, which could reduce clinical interpretability, hinder comparability, and introduce a risk of data-driven optimization. Our aim here is to evaluate radiomics derived from the clinically implemented high-uptake subvolume, maintaining fidelity to the protocol and the supporting evidence. For these reasons, we have retained the pre-specified threshold.
    In the revision, we clarify the rationale and provenance of the T/N > 2.0 cutoff and state explicitly that this threshold defined the escalated GTV in our dose-escalation cohort. We fully agree that threshold selection is an important topic and, as larger multiinstitutional datasets become available, we anticipate exploring generalizability in future work. We hope this explanation addresses the reviewer’s concern.
  7. Discuss further more the limitations because this excludes patients with varying IDH/MGMT status.
    Response (R1-C7): We thank the reviewer for highlighting this important limitation. In accordance with WHO CNS5 (2021) excluding IDH-mutant glioma from glioblastoma, our analysis focused on glioblastoma, IDH-wildtype (grade 4); cases with IDH mutation were therefore not included, and we now state this explicitly so that the scope of inference is clear. Regarding MGMT promoter methylation, our longitudinal design required adequate and consistent 18F-DOPA PET follow-up imaging to model remaining survival. From our previously reported (Ref. 16) full dose escalation cohort, patients with MGMT methylation typically had much longer survival than those who were MGMT unmethylated and 18F-DOPA PET follow-up imaging was often discontinued early due to standard of care progression calls, leaving too few complete longitudinal series to support reliable stratified modeling. Consequently, we did not perform MGMT-stratified analyses in this study.
    We acknowledge that excluding IDH-mutant tumors and not stratifying by MGMT limits generalizability across molecular subgroups. In the revision, we have expanded limitations in the discussion section to (i) clarify these inclusion choices and their rationale, and (ii) restrict our claims to IDH-wildtype MGMT-unmethylated glioblastoma.
  8. Although reported, the manuscript does not directly compare ROC AUC with other PET tracers, or conventional imaging models, which could inhibit perceived effect.
    Response (R1-C8): We thank the reviewer for this thoughtful comment. We agree that contextualizing AUC performance can aid interpretation; however, direct head-to-head comparison with other PET tracers or conventional imaging models is limited by the paucity of published post-treatment surveillance studies and by substantial heterogeneity in tracers, imaging protocols, endpoints, and uniquely for our cohort the use of 18F-DOPA-guided dose escalation, which likely alters the prevalence and timing of treatment effects (e.g., pseudoprogression) relative to standard-dose regimens.
    To provide context without over-interpreting across non-comparable cohorts, we have added a brief discussion comparing our results to the most closely related study we identified (Lohmann et al. (2020; Ref. 32)), which developed radiomics for distinguishing true progression from pseudoprogression after standard RT and reported an AUC of 0.74, modestly lower than our AUCs (0.81 for AutoGluon; 0.83 for EBM). We explicitly caution that differences in tracer protocols, treatment regimens, and outcome definitions preclude formal equivalence claims. The new text is included in revised manuscript.
    We hope this balanced comparison clarifies where our findings sit relative to prior work, while acknowledging that definitive cross-tracer or modality-level benchmarking will require harmonized, multi-center datasets: an effort we are actively pursuing.
  9. Talk about the performances of models in standard RT or mixed patient groups.
    Response (R1-C9): We thank the reviewer for this insightful suggestion. We agree that evaluating performance in standard RT cohorts and mixed patient groups would enhance the generalizability of our findings. At present, our model was developed in a dose-escalation cohort using 18F-DOPA FU imaging. Because the 18F-DOPA tracer is not FDA-approved for glioblastoma and is not reimbursed, its use in routine standard-dose RT settings at our institution has been limited, which constrained access to appropriate datasets for a direct analysis in standard RT or mixed groups.
    In the revision, we clarify this limitation explicitly restricting our claims to the studied cohort and emphasizing the exploratory/hypothesis-generating nature of the results. We also note our ongoing efforts to expand access through multiinstitutional collaboration and prospective studies, with the specific aim of testing the model in standard RT and more heterogeneous patient populations when such data becomes
    available. We hope this transparent framing addresses the reviewer’s concern while outlining a clear path toward broader validation.
  10. Mark which characteristics make predictions and discuss SHAP or other explainability techniques, along with EBM.
    Response (R1–C10): We thank the reviewer for this insightful suggestion. In the revision, we identify which characteristics drive predictions by reporting feature importance for both AutoGluon and EBM; The newly provided Table 3 summarizes feature importance derived in each model, and the Discussion contrasts the importance profiles across models.
    Regarding SHAP, we agree it is a valuable tool; however, applying it consistently here is non-trivial. AutoGluon uses a stacked, heterogeneous ensemble, for which a unified SHAP attribution requires custom, model-specific explainers for each layer and non-standard aggregation. For EBM, the model already provides exact additive contributions (shape functions), including explicit pairwise interactions that are not natively considered in SHAP. By design, EBM delivers per-feature and per-patient explanations aligned with the model’s structure. For these reasons, we relied on each model’s native explainability methods, which we believe satisfy the reviewer’s intent to clarify which characteristics drive the predictions.
  11. The quality of figure 1 needs to be enhanced.
    Response (R1-C11): We thank the reviewer for this observation. Figure 1 has been replaced with a higher-resolution version and enlarged fonts to enhance overall readability.
  12. Compare to baseline models.
    Response (R1–C12): We thank the reviewer for this helpful suggestion. In the revision, we implemented a baseline model using TBRmax (tumor-to-normal SUVmax), a commonly reported prognostic PET metric. The decision threshold was selected by grid search on the training set and then fixed and applied unchanged to the independent test set. This baseline achieved an AUC of 0.72 (training) and 0.66 (test), lower than the AutoML/AutoGluon and EBM models. We have added the baseline methodology to the Methods and its results to the Results, and we summarize the comparison in the Discussion.

Reviewer 2 Report

Comments and Suggestions for Authors
  1. Explain how missing data were handled in statistical analysis
  2. Report confidence intervals for AUC values (Autogluon and EBM), and other performance metrics like Brier score or decision-curve analysis to show clinical utility. Need to show how earlier detection would have altered management or outcomes 
  3. Provide feature count reduction flow in a table for clarity
  4. Only 35 patients with 140 total FU scans poses high risk of overfitting. Discuss measures to prevent overfitting given the small dataset. 
  5. Remaining survival is a novel but non-standard endpoint. The binary cut-off at 12 months seems arbitrary; justify with data (ROC-based Youden index or sensitivity analyses). 

Author Response

We express our heartfelt gratitude to both reviewers and the editors for their invaluable comments, which have significantly contributed to enhancing the quality of this manuscript. We also sincerely appreciate the editor’s kind consideration in granting us additional time, which allowed us to thoroughly address each of the reviewers’ suggestions in this revised submission. The manuscript has undergone thorough revisions in accordance with the provided feedback, and the responses to the reviewers' comments are summarized below. Within this document, responses are marked in blue for easy reference, and corresponding changes in the manuscript are highlighted in red and annotated with the respective reviewer comment identifier. For instance, "R1-C2" denotes the revision made in response to the second comment (C2) from Reviewer 1 (R1).

  1. Explain how missing data were handled in statistical analysis
    Response (R2-C1): We thank the reviewer for this helpful comment. We have now explicitly described our missing data handling in Methods section. Briefly: (i) because this is an image-based study, if a follow-up PET scan was absent at a given timepoint, that timepoint was omitted from longitudinal analyses (no imputation); (ii) to maintain a homogeneous cohort of glioblastoma with unmethylated MGMT, patients without a determined MGMT status were excluded; and (iii) when a scan contained no voxels exceeding the predefined T/N > 2.0 threshold, the high-uptake ROI was empty and its radiomic features were set to zero by design, reflecting the absence of above-threshold uptake. No other imputation procedures were applied.
  2. Report confidence intervals for AUC values (Autogluon and EBM), and other performance metrics like Brier score or decision-curve analysis to show clinical utility. Need to show how earlier detection would have altered management or outcomes.
    Response (R2-C2): We appreciate the reviewer’s thoughtful suggestions. In the revision, we now report 95% confidence intervals for the AUCs of both AutoGluon and EBM, estimated via 2,000 bootstrap resamples, and we include these in the Results. To address clinical utility, we performed a decision curve analysis (DCA) on the independent test set (now Figure 6 (a)). Both models demonstrate positive net benefit over “treat-all” and “treat-none” strategies across clinically plausible thresholds, supporting potential value for guiding surveillance or intervention. Finally, we expanded the Discussion to explain how earlier detection
    could alter management in practice: prompting earlier confirmatory imaging and consideration of salvage therapies (e.g., re-resection, re-irradiation, systemic therapy, or trial enrollment). Consistent with this, our cohort-level analyses show earlier identification than RANO in a substantial subset, aligning with the intended clinical use.
  3. Provide feature count reduction flow in a table for clarity
    Response (R2-C3): We appreciate the reviewer’s helpful suggestion. In the newly provided supplementary materials document, we have added Supplementary Table S3 detailing the feature count-reduction flow. We also inserted a concise summary in Section 3.2 (Radiomics Feature Selection) of the revised manuscript: 115 features were initially extracted; 104 were retained after univariable predictivity screening; and 47 were selected following cross-validation (11 shape, 12 first-order, 24 texture). Category-based pair-correlation filtering reduced the number of selected features to 6. Finally, manifold learning reduced these to two abstracted features for modeling.
  4. Only 35 patients with 140 total FU scans poses high risk of overfitting. Discuss measures to prevent overfitting given the small dataset.
    Response (R2-C4): We sincerely thank the reviewer for pointing out this limitation. We acknowledge that the small sample size is a key limitation of our study, and we address this in the forefront of the discussion section. In the revision, we have (i) explicitly framed our analyses as exploratory/hypothesis-generating, and (ii) tempered the Abstract (Conclusions) and Discussion language to avoid over-interpretation.
    There are practical reasons for the cohort size that we now state more clearly. Glioblastoma is a relatively uncommon tumor, which often results in small sample sizes, particularly when exploring novel imaging techniques. 18F-DOPA is not yet FDA-approved for glioblastoma; therefore, access is constrained by research funding and institutional resources. In addition, to our knowledge this is a unique clinical dose-escalation trial guided by 18F-DOPA PET, making accrual intrinsically challenging. We have clarified these points in the limitation paragraph of Discussion.
    Methodologically, we took steps to mitigate overfitting given the sample size: we restricted model complexity through a multi-stage approach (initial univariable screening of radiomic features with cross validation followed by manifold learning to reduce the feature space and degrees of freedom before modeling). This strategy is intended to improve robustness in small-sample settings.
    Finally, we have added text outlining our ongoing plans to enable broader validation: a new prospective study of additional patients for 18F-DOPA PET guided radiotherapy is underway, and we are actively pursuing multiinstitutional collaboration to expand the cohort and perform external validation. We have also included contextual references noting that several prior radiomics studies in glioblastoma (e.g., Ref. 32, n=34) were necessarily small yet still generated clinically informative hypotheses; we cite these to contextualize feasibility rather than to justify generalization. We hope these clarifications and revisions address the reviewer’s concern and transparently convey the scope and limitations of our current findings.
  5. Remaining survival is a novel but non-standard endpoint. The binary cut-off at 12 months seems arbitrary; justify with data (ROC-based Youden index or sensitivity analyses).
    Response (R2-C5): We thank the reviewer for this thoughtful comment. We agree that remaining survival (RS) is a non-standard endpoint; we use it as a pragmatic, clinically accessible surrogate to complement (not replace) formal progression determinations, which often lack serial pathological confirmation in post-treatment glioblastoma surveillance.
    The 12-month cutoff was chosen a priori based on published reports (Ref. 11, 26, 32) indicating median survival of approximately one year after true progression in glioblastoma, and because it aligns with a clinically meaningful decision horizon. To address the reviewer’s concern about arbitrariness, we conducted sensitivity analyses re-defining high-risk transition at RS ≤10 and RS ≤14 months. Model performance was stable across these thresholds, with negligible changes in AUC, supporting the robustness of our findings. This analysis is now included in the Results section of the revised manuscript.
    We note that a ROC/Youden index approach is most appropriate for selecting a classifier threshold on a predictor within a given dataset; applying it here to choose a clinical time horizon could introduce circularity and reduce external interpretability. For this reason, we favored a literature-informed threshold supplemented by prospective sensitivity checks. We have clarified these points in the Discussion section regarding its limitations and clinical interpretation.

Round 2

Reviewer 1 Report

Comments and Suggestions for Authors
  1. Highlight limitations of your work clearly.
  2. Improve quality of all Figures.

Author Response

We express our heartfelt gratitude to the reviewer in the second round for the suggestions to improve the quality of our manuscript. Within this document, responses are marked in blue for easy reference, and corresponding changes in the manuscript are highlighted in red.

Comment 1: Highlight limitations of your work clearly.

Response: We sincerely thank the reviewer for prompting a clear articulation of our study’s limitations. The Discussion already concludes with a comprehensive paragraph detailing the key constraints of this work, including the small dataset size, single-institution data from a unique clinical trial, and our focus on a single set of pathological biomarkers. Throughout, we emphasize that the study is exploratory/hypothesis-generating, and that generalization of these findings will require validation in larger, more diverse, ideally external datasets. In the revision, we further strengthened the Abstract and Introduction to underscore this exploratory framing as early as possible to the readers and to help prevent over-interpretation or over-generalization. We hope these clarifications meet the reviewer’s intent regarding the presentation of study limitations.

Comment 2: Improve quality of all Figures.

Response: We truly thank the reviewer for this helpful suggestion. Interpreting the comment as primarily concerning resolution and legibility, we have replaced all figures with high-resolution TIFF files and verified readability at the journal’s column width. All the TIFF files are also uploaded separately from the manuscript for editorial use. If the reviewer had additional aspects in mind, we would be grateful for guidance and will incorporate it promptly.
